# Chemical and Biological Properties of Three Poorly Studied Species of *Lycium* Genus—Short Review

**DOI:** 10.3390/metabo12121265

**Published:** 2022-12-15

**Authors:** Maria da Graça Miguel

**Affiliations:** Departamento de Química e Farmácia, Mediterranean Institute for Agriculture, Environment and Development, Faculdade de Ciências e Tecnologia, Universidade do Algarve, Campus de Gambelas, 8005-139 Faro, Portugal; mgmiguel@ualg.pt

**Keywords:** biological properties of *Lycium genus*, chemical properties of *Lycium genus*, *Lycium barbarum*, *Lycium europaeum*, *Lycium intricatum*, *Lycium schweinfurthii*

## Abstract

The genus *Lycium* belongs to the Solanaceae family and comprises more than 90 species distributed by diverse continents. *Lycium barbarum* is by far the most studied and has been advertised as a “superfood” with healthy properties. In contrast, there are some *Lycium* species which have been poorly studied, although used by native populations. *L. europaeum*, *L. intricatum* and *L. schweinfurthii,* found particularly in the Mediterranean region, are examples of scarcely investigated species. The chemical composition and the biological properties of these species were reviewed. The biological properties of *L. barbarum* fruits are mainly attributed to polysaccharides, particularly complex glycoproteins with different compositions. Studies regarding these metabolites are practically absent in *L. europaeum*, *L. intricatum* and *L. schweinfurthii*. The metabolites isolated and identified belong mainly to polyphenols, fatty acids, polysaccharides, carotenoids, sterols, terpenoids, tocopherols, and alkaloids (*L. europaeum*); phenolic acids, lignans, flavonoids, polyketides, glycosides, terpenoids, tyramine derivatives among other few compounds (*L. schweinfurthii*), and esters of phenolic acids, glycosides, fatty acids, terpenoids/phytosterols, among other few compounds (*L. intricatum*). The biological properties (antioxidant, anti-inflammatory and cytotoxic against some cancer cell lines) found for these species were attributed to some metabolites belonging to those compound groups. Results of the study concluded that investigations concerning *L. europaeum*, *L. intricatum* and *L. schweinfurthii* are scarce, in contrast to *L. barbarum*.

## 1. Introduction

The botanical definition for berries is fleshy fruit produced from a single ovary. Nevertheless. berries are generally known as small edible fruits brightly colored in shades of red, blue, and purple and without a pit, although they may contain seeds, such as strawberries, blueberries, blackberries, raspberries, wolfberries or goji berries, among others. These colors are usually attributed to the presence of anthocyanins. However, not all berries contain anthocyanins, such as wolfberries. Only blueberries are true berries, according to the botanical definition [1].

The genus *Lycium* belongs to the Solanaceae family, and comprises about 97 species and six varieties, which are distributed in South America (32 species), North America (24 species), South Africa (24 species), and temperate Europe and Asia (12), and two occur in Eurasia as well as in Africa [2]. In China, there are seven species and three taxonomic varieties, mostly distributed in Gansu, Qinghai Provinces, Xinjiang, and Ningxia Autonomous regions [3]. According to Fukuda et al. [4], the *Lycium* origin is not yet known; nevertheless, it was hypothesized that the genus originated somewhere in South or North America, and then dispersed to southern Africa. In turn, the same authors also hypothesized that the Eurasian and Australian species of *Lycium* originated from southern Africa.

*Lycium* species are shrubs or small trees that can be found in the arid and semi-arid regions of North and South America, Africa, and Eurasia. The species belonging to this genus often show thorns on the stem and leaves. Generally, they can be differentiated by taking into account the form and size of leaves, the corolla and stamen lengths, the thorn on the stem, color and taste of the fruit, and size and number of seeds [2].

Currently, *Lycium barbarum* L. has been advertised as a “superfood” with healthy properties [2], being one of the most studied species of the *Lycium* genus. For this reason, only a very short review will be made on this species in the present section. *Lycium barbarum* L. is a perennial deciduous shrub that can be found in arid and semi-arid regions of northwestern China, including Xinjiang Province, Qinghai Province, Ningxia Province, Gansu Province, and other provinces, and in southeastern Europe and Mediterranean areas [5,6]. The fruit of this species is known as wolfberry, maybe by the fact that Chinese farmers observed wolves eating these berries. This fruit is also named goji berry, which is orange-red in color, ellipsoid, approximately 2 cm deep, and with a sweet-and-tangy flavor [7]. These edible fruits have been largely used in traditional Chinese medicine for centuries, and as a functional food [5,6]. It is listed in the Traditional Chinese Pharmacopeia [6]. It is also traditionally used in Korean, Japanese, and Vietnamese medicine [8].

The Ningxia Hui autonomous region of China is considered the global birthplace of the wolfberry, cultivated centuries ago; nevertheless, the number of production regions has been increasing, so producers intend to possess tools that permit to distinguish the quality of fruits according to their distinct geographical origin. This will determine the price of the product becoming consequently of added value to the producers [9,10]. In spite of the subjectivity of the methods, it is frequent to discriminate the wolfberries through color, shape, odor, taste, and other qualities highly dependent on the personal elicited sensorial impressions. For this reason, several methods have been developed to overcome this subjective factor, such as voltametric electronic tongue [10]; electronic tongue (coupled combined with deep learning algorithm (convolutional neural network) [11]; mineral profile with chemometric approaches [12]; two-dimensional correlation spectroscopy combined with deep learning algorithm, such as convolutional neural network [13]; surface near infrared spectra combined with multi-class support vector machine [14]; Fourier transform infrared spectroscopy combined with artificial neural networks [15]; carbon isotope analysis of specific volatile compounds (e.g., limonene, tetramethylpyrazine, safranal, geranyl acetone, and β-ionone) by gas chromatography-combustion isotope ratio mass spectrometry with headspace-solid phase microextraction [8]; stable isotopes, earth elements, free amino acids, and saccharides using orthogonal projection to latent structure-discriminant analysis [16]; non-targeted liquid chromatography coupled to quadrupole time-of-flight mass spectrometry combined with statistical analysis [17] are some examples.

China has the largest cultivation or production area of wolfberries in the world, and the fruits from Ningxia recognized as top quality. Wolfberries from Zhongning County can be sold at high prices [16]. The importance of this culture has led to the development mechanized harvesting technologies that accelerate the harvesting process without damaging the fruits, and robots able to replace manual labor. This will increase margins and profits [18,19,20].

Fresh wolfberries have high water content and tender tissue, therefore, deteriorate quickly after harvesting, by microbial attack and mechanical damage [21]. In this context, it is necessary to find strategies to increase shelf life of fresh fruits without losing quality. For example, Ban et al. [21] reported that the combination of pre-storage heat treatment and a chitosan coating lowers decay compared with untreated wolfberry fruits, and maintains higher levels of ascorbic acid, total phenolic contents and antioxidant capacity. More recently, Xiang et al. [22] observed that the combination of salicylic acid and hypoxia (controlled and modified atmosphere conditions) could synergistically suppress respiration rate and antagonistically maintain ascorbic acid and color quality, while salicylic acid could alleviate physiological disorder. Liang et al. [8] found that 10% CO_2_ better maintained the physiological quality and conferred the reduction in weight loss of wolfberries, decay index, color change, and antioxidant activity enhancement. The preservation of wolfberries also involves the utilization of some microorganisms (e.g., *Bacillus subtilis* strain CL2) which are able to release volatile organic compounds, particularly 2,3-butanedione and 3-methylbutyric acid, that significantly reduced the weight loss rate of wolfberry fruits caused by the four pathogenic fungi (*Mucor circinelloides* LB1, *Fusarium arcuatisporum* LB5, *Alternaria iridiaustralis* LB7, and *Colletotrichum fioriniae* LB8) [23].

In spite of diverse postharvest methodologies having been developed to increase the fresh wolfberry shelf-life; the fact is that the dried berries are more popular, since they can be preserved for longer periods with minimal chemical deterioration and microbial spoilage. In addition to these, there is also a reduction in the volume of the product that is better for transport and storage costs. Solar drying or hot air have been applied to Chinese wolfberries, nevertheless these methods may cause quality losses [24]. In this context, new drying technologies have been tried and/or improved keeping in mind the lowest possible energy consumption during the process. Some examples include electrohydrodynamic system [24,25], electrohydrodynamic system combined with oven drying [26], electrohydrodynamic drying in a multiple needle-to-plate electrode system [27], pulsed vacuum drying [28], pulsed vacuum drying combined with carboxymethyl cellulose coating pretreatment [29], solar assisted heat pump drying system [30], far-infrared radiation heating assisted pulsed vacuum drying [31], low-intensity pulsed ultrasound [32] and radio frequency-hot air drying process [33]. Cui et al. [34] in a recent review article concluded that infrared drying and microwave drying can be combined with other drying methods to improve drying efficiency. In addition, the authors [34] also reported that freeze drying, pulsed vacuum drying, and electro-hydrodynamic drying, can be useful to maintain the appearance and nutrients as well as the higher rehydration ratio.

Numerous compounds have been reported for wolfberries, nevertheless polysaccharides are the most significant and several biological properties have been attributed to these metabolites, namely antioxidant, hepatoprotective, anti-inflammatory, cardioprotective effect, hypoglycemic and immune activities, among other biological attributes [35,36,37]. They are generally complex glycoproteins with different composition, although the monosaccharides are generally the same (rhamnose, arabinose, mannose, xylose, galacturonic, glucose, and galacturonic acid) [38]. The polysaccharide complex, with an approximate molecular weight (MW) of 10–2300 kDa, is mostly composed of (1→3)-β-D-galactopyranosyl, (1→6)-β-D-galactopyranosyl, and (1→4)-α-D-galactopyranosyluronic acid residues. A glycan-*O*-Ser glycopeptide structure has been mostly considered for the efficacy of *L. barbarum* as well as 2-*O*-(β-D-glucopyranosyl) ascorbic acid (could serve as a stable vitamin C substitute) [36,39,40]. Carotenoids are the second major group of metabolites of wolfberries (zeaxanthin, β-cryptoxanthin, β-carotene, and mutatoxanthin, among other minor carotenoids) [40]. Phenolic acids, flavonoids, phenylpropanoids, coumarins, lignans and their derivatives, monoterpenes (phellandrene, sabinene, terpinene), alkaloids and spermidine alkaloids [38,41,42] are secondary metabolites that can be found in wolfberries, as well as vitamins. Wolfberries are also a source of dietary fibers, some minerals (K, Cu, Mn, Fe, Zn), fatty acids (linoleic, oleic, palmitic, and stearic acids), amino acids (e.g., proline and serine) and non-protein amino acids (e.g., γ-aminobutyric acid, hydroxyproline, and citrulline) [43]. The chemical profile of these berries is dependent on diverse factors, such as environmental conditions, pre- and post-harvest factors, rhizosphere bacterial community structure and genetic heritage [43,44].

The biological properties of wolfberries have been also largely reported and reviewed. For instance, when performing a browse search in the Web of Science platform in November 2022 using the following key words “*Lycium barbarum* review”, and restricting for year 2022 only, it is possible to find a large number of review papers that have in their scope of analysis this species, as well as its antioxidant, anti-inflammatory, anti-aging, hypoglycemic and hypolipidemic activities, modulation of gut microbiota, neuroprotective, neuroprotective effects on retinal ganglion cells, immunomodulatory, positive effects on cognitive impairment, anti-fatigue effect, hepatoprotective, and anti-tumor effect [45,46,47,48,49,50,51,52,53,54,55,56,57,58,59,60,61,62]. The role of the plant polysaccharides, including that of *L. barbarum*, in drug delivery system in tumor treatment, targeted therapy, and wound healing was also reviewed and discussed [63]. In these review articles, the mechanisms involved in the biological activities were also addressed and discussed. There are many scientific studies involving the chemical composition, biological properties, conservation, and drying methods of the *L. barbarum* fruit, resulting in more than 20 review articles in indexed scientific journals, in the first 11 months of 2022. The same cannot be observed for other species, such as *L. europaeum*, *L. intricatum* and *L. schweinfurthii*. This review, by compiling the little information that exists on 3 species of *Lycium*, can contribute to arouse interest in such species as possible sources of food, food supplements or even medicines. As such, the main goal/purpose of this revision study is to share more information and understand the relevance of the less studied *Lycium* species.

## 2. Other *Lycium* Species

According to the review made by Yao et al. [2], there are four species of *Lycium* that can be found in Portugal, although also in other regions of Europe, Asia, America and Africa. One example is *L. europaeum* L. that can be found in Portugal, Spain, France, Israel, Palestinian Territory, India, Algeria, Tunisia, and Egypt. The berries and young shots are used as food, and the fruit, leaves, bark, and whole plant as folk medicine. But in the same review Yao et al. [2] also cited *L. infaustum* Miers that grows in several countries of South America (Argentina, Colombia, Bolivia, Ecuador, Peru, Paraguay), Central America (Dominican, Turks and Caicos Islands, and Jamaica), and Portugal. The application of this species as food or medicine is unknown and up until then no record could be found, thereby making any type of revision unfeasible in the present review article. The application of *L. intricatum* Boiss as food was also unknown, but there were records of it being used for eye treatment (fruits) and its seeds as helminthisis and digestive. This species was registered in Portugal, Spain, Italy, Morocco, Algeria, Tunisia, Egypt, Mauritania, Saudi Arabia, and Mexico [2]. *L. schweinfurthii* Dammer is aother species reported by Yao et al. [2] as being detected in Portugal but also in Spain, Israel, Morocco, Greece, Algeria, Egypt, Tunisia, Mauritania, and Cyprus. Only the leaf and fruit are used in the case of stomach ulcer, according to the same review [2].

### 2.1. Lycium europaeum L.

*Lycium europaeum* is a spiny shrub, 1–4 m tall, with red fruits (Figure 1), oblanceolate leaves (20–50 × 3–10 mm), calyx (2–3 mm) 5-dentate or 2-lipped, corol1a narrowly infundibuliform (11–13 mm), pink or white, with stamens usually exserted. Flowers can be solitary or in clusters of two [2].

#### 2.1.1. Chemical Composition and Biological Properties

In a review made by Wannes and Tounsi [64] published in 2021, the chemical composition of *L. europaeum* was addressed and the authors concluded that only 30 constituents were identified, distributed by the following groups: polyphenols, fatty acids, polysaccharides, carotenoids, sterols, terpenoids, tocopherols, and alkaloids (Figure 2). Nevertheless, in a previous review article [65], one pentacyclic triterpenoid (ursolic acid) in root bark, and four steroids were also reported in *L. europaeum* (cycloartenol, stigmasterol and lanosterol in roots, and β-sitosterol in aerial parts) (Figure 2). The authors [66] found diverse pharmacological properties of the same species (antioxidant, antinociceptive, hepatoprotective, nephroprotective, hypolipidemic, and cytotoxicity activities). Nevertheless, a correspondence between the constituents identified and the biological properties was not established. For this reason, the authors concluded the necessity of this approach as well as the need to determine the mechanisms involved for each bioactive compound, particularly the molecular targets and pathways controlling the biological activities. Anti-inflammatory activity was also found by Akter et al. [66] in a review study. Due to these recent published articles on this species of the past year (2021), the present research is limited to the year of 2022, up to November. During this period there were no appreciable advances and the records found included: (a) ethnobotanical studies; (b) detection for the first time of an infection by a specific fungus of *Lycium* species in Iran; (c) in vitro antioxidant, anti-acetylcholinesterase, anti-butyrylcholinesterase and anti-urease activities of crude extracts and their fractions; and d) antioxidant and anti-inflammatory activities of fruit methanolic extracts.

The chemical and digestibility qualities of preferred forage species by lactating Somali camels in Kenya were studied. *L. europaeum* was one of the most preferred with 208 bite counts being found (the highest count), during the dry season. The dry matter was 20.5%, the crude protein 25.7%, the ash concentration 22.9%, the calcium concentration 3.4%, whereas the percentage for phosphorous was 0.3%. The in vitro dry matter digestibility was 81.6%, associated with the relatively low neutral detergent fiber (34.2%). Therefore, these values associated with the relative high crude protein makes *L. europaeum* a quality and clearly beneficial forage with a positive effect on overall camel performance [67].

The infusion of *L. europaeum* leaves provided orally is used in the treatment of endo- and ectoparasitic infestations in sheep in the Rif region, north of Morocco. However, the authors [68] indicate that this finding possesses a relatively low fidelity level (58%). Fidelity level is defined as the percentage of informants who claim the use of a specific species for the same major use.

Mollel et al. [69] in an ethnobotanical study in Tanzania, and in a total of 147 species in 16 genera, found records of the use of *L. europaeum* barks in the treatment of syphilis and cancer with a low relative frequency of citation (0.18) and also a low use-value (0.02). Such results indicate a very low confidence for this species in the treatment of those illnesses. *L. europaeum* is also recorded as medicinal plant in another ethnobotanical study, this one from eastern Libya, but for the treatment of rheumatic, constipation, wounds, and dermatitis. The relative frequency of citation and/or fidelity level was not reported [70]. *L. europaeum* was for the first time reported in the Wadi Kaam in northwest Libya (130 km east of Tripoli), although only four individuals were found [71].

The antioxidant activity of ethanol extracts and their fractions (chloroform, ethyl acetate, *n*-butanol and aqueous) was evaluated through the capacity for scavenging some free radicals such as 1,1-diphenyl-2-picrylhydrazyl (DPPH), 2,2′-azino-bis-3-ethylbenzthiazoline-6-sulphonic acid (ABTS), and superoxide anion radicals. Other evaluation methods were also used, such as the β-carotene bleaching assay, cupric reducing and ferric reducing activities. The capacity of the extracts and fractions for inhibiting the enzyme acetylcholinesterase, butyrylcholinesterase and urease activities was also assessed. The *n*-butanol fraction from roots had the highest antioxidant activity in all the assays, and was also the most active against acetylcholinesterase, butyrylcholinesterase and urease activities (IC_50_ values of 92.63, 118.26 and 135.60 μg/mL, respectively). There was a correlation between the activities found and the amounts of total phenols, flavonoids, and tannins. According to the authors, roots of *L. europaeum* are good candidate to be explored as a source of bioactive products [72].

Methanol and ethyl acetate were the most adequate solvents for extracting the phenolic fraction of *Lycium europaeum* fruits that showed the highest antioxidant and anti-inflammatory activities. The methanol extract of *L. europaeum* fruits was the richest in phenolic compounds, particularly and by descending order, ferulic acid, catechin, naringenin, apigenin and gallic acid. These extracts were not effective in the inhibition of cancerous human lung carcinoma A-549, colon adenocarcinoma DLD-1 and non-cancerous human skin fibroblast WS-1 growth [73]. Although the absence of activity against these cancer cells or non-cancerous WS-1 cells, the methanolic extract had the highest capacity for scavenging the DPPH free radicals and reducing power capacity along with the water extract, nevertheless the ethyl acetate extract had better capacity for scavenging ABTS free radicals. The ex vivo studies, using the dichlorofluorescein-diacetate in a cellular-based assay showed that ethyl acetate and methanol extracts were the most promising samples. According to the authors [73] only the DPPH assay did not correlate with the phenol or flavonoid concentrations of the extracts. In this case, the authors [73] suggested that polysaccharides, alkaloids and terpenoids have an important role on the antioxidant activity along with the phenols. The same authors conclude that such results may indicate the possible utilization of *L. europaeum* fruits as a source of bioactive molecules or products for pharmaceutical industry.

#### 2.1.2. Fungal Infection

*Arthrocladiella mougeotii* (Ascomycota, Helotiales) is the only fungus species that infects various species of *Lycium* worldwide in places such as Russia, Europe, China, Japan, New Zealand, USA, Turkey and Israel. For the first time, the infection of *L. europaeum* by *Arthrocladiella mougeotii* was reported in Iran. In the same work, the authors revealed that this infection was previously reported in other plant species (*Deutzia gracilis*), albeit after reexamination of the plant material and compared with type material of *D. gracilis* and two authentic herbarium specimens of *L. europaeum*, in the JSTOR Global Plants database, the authors concluded that there had been a misidentification. With this finding, the authors reported an infection of *L. europaeum* by *Arthrocladiella mougeotii* for the first time, in Iran [74].

### 2.2. Lycium schweinfurthii Dammer

According to the last review found in the present research, Yao et al. [2] reported that leaves and fruits of *L. schweinfurthii* are used for stomach ulcer. More recently, Ajjoun et al. [75] reported that in Morocco, the decoction of the whole plant is used for hair care; nevertheless, after consulting the original source of information [76], that application is attributed to *L. intricatum* Boiss. Ajjoun et al. [75] considered *L. intricatum* as synonymous with *L. schweinfurthii*. This fact may lead to the consideration that *L. intricatum* is the species used as hair care and not *L. schweinfurthii*.

In fact, morphologically, *L. schweinfurthii* (Figure 3) and *L. intricatum* (Figure 4) are quite similar. They have flowers in which the floral tube is relatively long, lobes and the calyx are quite short, nevertheless the calyx in *L. schweinfurthii* is shorter and often as long as it is broad. Furthermore, the corolla lobes in *L. schweinfurthii* have rather evident ribs or are more or less whitened while those of *L. intricatum* are lilac. Concerning the leaves, they are frequently much more succulent in *L. intricatum* with an almost cylindrical shape whereas in *L. schweinfurthii* they are mostly flat. Finally, the color of the ripe fruit of *L. schweinfurthii* is black, whereas in *L. intricatum*, the color of ripe fruit is red [77]. These authors claimed that it may not be entirely improbable that at least some of the reports of *L. intricatum* in Italy could be related to the *L. schweinfurthii*, thereby it will be necessary to verify the identity of plants previously attributed to *L. intricatum*. Therefore, this species may occur in Italy, although until now, it is described as non-existent in this country, as reported in the review article of Yao et al. [2].

Although Yao et al. [2] had not reported the existence of *L. schweinfurthii* in Palestine, the same authors reported the biological activities of this species collected in Til village, Nablus (Palestine), a research work of Jamous et al. [78], as well as its utilization for stomach ulcer from plants collected in different localities of Gaza Strip (Palestine) [79]. More recently, Ighbareyeh et al. [80] also found *L. schweinfurthii* in Palestine, particularly in the region of Beit Jibrin.

#### 2.2.1. Chemical Composition and Biological Properties

The chemical composition of diverse parts of *L. schweinfurthii* is scarce. β-Sitosterol, rutin, diosgenin (Figure 5) were previously reported in the aerial parts of this species in Egypt [81]. Ewais et al. [82] reported the presence of flavonoids, glycosides, phenolic acids in fresh roots, stems, leaves, and flowers collected in Egypt. By using chromatographic techniques, five compounds including quercetin, kaempferol, apigenin, gallic, and ferulic acids, were separated from leaves. These compounds were characterized by ultraviolet (UV), proton-nuclear magnetic resonance (^1^H-NMR) and mass spectrometry (MS). Later on, Elbermawi et al. [83] found a new glycoside (3-methoxy-4-*O*-β-D-glucopyranosyl-methylbenzoate) along with diosmetin, luteolin, quercetin, diosmetin-7-*O*-β-D-glucoside, and 3-hydroxy-1-(4-hydroxy-3-methoxyphenyl)-2[4-(3-hydroxy-1-(*E*)-propenyl)-2-methoxyphenoxy]-propyl-β-D-glucopyranoside (Figure 5), present in ethyl acetate fraction, obtained from fractionation of the aerial part of the methanolic extract. Their structures were elucidated by 1D and 2D NMR as well as MS analysis. The new glycoside compound identified by the authors had a potent α-glucosidase inhibitory activity as well as diosmetin, luteolin, quercetin and 3-methoxy-4-*O*-β-D-glucopyranosyl-methylbenzoate (Table 1). According to the authors, 3-methoxy-4-*O*-β-D-glucopyranosyl-methylbenzoate is more stable in vivo than the flavonoids that have similar activities. These compounds can be easily degraded within the body into the corresponding phenylacetic acid, with consequent loss of activity. Furthermore, the same group in other work [84] isolated more than 28 compounds in all fractions of the same crude methanolic extract. Moreover, the authors evaluated the cytotoxic potential of the isolates in skin cancer (G-361) and colon cancer (HCT-116 and CaCo-2) cell lines (Table 1).

The compounds identified by Elbermawi et al. [84] were: vaginatin, 11*S*-methoxy-11,12-dihydrophytuberin, (*E*)-docosanoyl ferulate and (*Z*)-docosanoyl ferulate, β-sitosterol and β-sitosterol-3-*O*-β-D-glucoside, 9*S*-methoxy-benzocyclononan-7-one, isoscopoletin, pinoresinol*, trans*-cinnamoyl tyramine, *trans*-ferulyl tyramine, *trans*-ferulyl-3-methoxytyramine, diosmetin, apigenin, luteolin, *p*-hydroxybenzoic acid, vanillic acid, kampferol, quercetin, methyl-α-D-fructofuranoside, meliasendanin D, *trans*-ferulic acid, *p*-coumaric acid, diosmetin-7-*O*-β-D-glucoside, quercetin-3-*O*-β-D-glucoside-6-α-D-rhamnoside (rutin), 3-methoxy-4-*O*-β-D-glucopyranosyl-methylbenzoate, gallic acid, and 3-hydroxy-1-(4-hydroxy-3-methoxyphenyl)-2[4-(3-hydroxy-1-(*E*)-propenyl)-2-methoxyphenoxy]-propyl-β-D-glucopyranoside (Figure 5). Diosgenin reported previously [81] was not found by the authors [84]. 11*S*-Methoxy-11,12-dihydrophytuberin and 9*S*-methoxy-benzocyclononan-7-one were new natural products. Concerning the cytotoxicity of these compounds on a skin cancer (G-361) cell line, the authors concluded that (*Z*)-docosanoyl ferulate and meliasendanin were not cytotoxic against G-361 cells in contrast to the remaining compounds, which presented potent cytotoxic effects, particularly diosmetin, kaempferol, gallic acid and vaginatin (Table 1). Regarding colon cancer HCT-116 cells, apigenin was the most effective. Apigenin and diosmetin were the most active against the colon cancer CaCo-2-cells (Table 1). All these positive results also have the advantage that these anticancer compounds showed minimal cytotoxicity towards normal cells. Nevertheless, the authors also found that the dichloromethane extract was much more toxic towards G-361 cell line than the isolated compounds, indicating a possible synergism effect among the constituents present in the extract [84]. These results may contribute to the possible application of this species as a supplement or even as a medicine, after adequate studies, and not only for use as fuel after cutting, according to the study made by Bedair et al. [85], in Egypt.

#### 2.2.2. Biotechnological Production of Secondary Metabolites

Taking into account the secondary metabolites previously found by diverse authors for the Egyptian *L. schweinfurthii* [83,84,85,86], Mamdouh and Smetanska [86] aimed to obtain *callus* and cell suspension cultures of this species for use in bio-factories for secondary metabolites production. This approach could provide a cost-effective alternative to traditional cultivation, although the successfully cases have been few [87]. Mamdouh and Smetanska [86] established the best conditions of the culture growth and subsequently for the production of the total phenols and flavonoids and antioxidant activity (capacity for scavenging DPPH and ABTS free radicals).

For the optimization, diverse factors were evaluated such as plant growth regulators and their combinations in the Murashige and Skoog medium (MS) for *calli* and N2 for cell suspension culture. In the suspensions, the authors studied the effect of diverse concentrations of sucrose on the growth and secondary metabolites production. For *calli*, the best conditions for growth were not the same as those for the production of phenols and flavonoids: the best medium for *callus* multiplication was MS medium fortified with 1 mg/L of both 2,4-dichlorophenoxy acetic acid (2,4-D) and 1-naphthyl acetic acid (NAA), whereas for secondary metabolites, the best combination was MS fortified with 2 mg/L NAA [86].

The suspension cultures in the N2 medium supplemented with 30 g/L induced better cell multiplication than lower sucrose concentrations, although for the production of metabolites as well as for the antioxidant activity, the best sucrose concentrations were 5 and 30 mg/L. According to the authors [86], such results can be useful in the large-scale production of phenols with antioxidant activity. Another biotechnological way to produce secondary metabolites is through in vitro micropropagated plants, and this approach was also made by the same group [88], who developed a protocol for in vitro micropropagation of *L. schweinfurthii* with genetic stability, assessed through a random amplified polymorphic DNA (RAPD), inter-simple sequence repeats (ISSR), and one biochemical technique, sodium dodecyl sulfate-polyacrylamide gel electrophoresis (SDS-PAGE), able to scavenge free radicals and produce phenols. The leaves of the micropropagated plants produced phenols, measured as total phenols and total flavonoids, and one phenolic acid (ferulic acid) was identified by high performance thin-layer chromatography (HPTLC). The capacity for scavenging the DPPH and ABTS free radicals were IC_50_ = 0.43 and 1.99 mg/mL, respectively. As aforementioned, the authors [88] considered that micropropagated plants of *L. schweinfurthii* can be another way to obtain plant material for in vitro secondary metabolite production. Nevertheless, nothing is reported about the production of clones with the best capacity for accumulating the secondary metabolites with antioxidant activity. Moreover, no reference to the production of clones for further field growing is given and, therefore, the objective to obtain micropropagated plants is not clear.

#### 2.2.3. Secondary Metabolites from Endophytic Fungus

More recently, Elbermawi et al. [89] isolated from the fresh leaves of *L. schweinfurthii*, collected in the International Coastal Road, 40 km west from Gamasa City (Egypt), an endophytic fungus identified as *Alternaria* sp., which was then made to grow on solid rice culture media. From the ethyl acetate extract of this culture media, the authors isolated and identified six phenolic compounds (talaroflavone, alternarienoic acid, altenuene, altenusin, alternariol and alternariol-5-*O*-methyl ether) (Figure 6). Alternarienoic acid, altenuene and altenusin had potent in vitro α-glucosidase inhibitory activity with IC_50_ values of 7.95, 40.38 and 46.14 μM, respectively, much better than the positive control (acarbose) which IC_50_ value was 283 μM (Table 2). The pancreatic lipase inhibitory activity also assayed by the authors, showed that these three compounds also had the highest inhibitory activity, particularly altenuene, which presented a IC_50_ value of 3.18 μM (Table 2), much closer to the IC_50_ value (1.35 μM) of the positive control (orlistat) than the remaining two samples (20.82 and 21.46 μM, for alternarienoic acid and altenusin, respectively [89]. The molecular docking study was done to predict the preferred fitting between two of the interacting chemical moieties of the phenolic compound and protein, using a computational simulation using through the Molecular Operating Environment package (MOE). According to this approach, the authors concluded that alternarienoic acid, altenuene and altenusin showed a maximum number of interactions with amino acids residues in the active site were able to maintain interaction with essential key residues, later confirmed through protein ligand interaction fingerprints [89]. According to the authors, these inhibitory activities can be promising naturally occurring anti-diabetic candidates. In spite of this conclusion, it is also important to stress that some of these phenolic polyketides are mycotoxins and they present genetic, reproductive, and developmental toxicity, and therefore may lead to adverse effects on health [90]; as such, those type of conclusions should be considered with care.

### 2.3. Lycium intricatum Boiss

According to the compilation made by Lombardo et al. [94], *L. intricatum* has two subspecies with different distribution: *L. intricatum* subsp. *pujosii* Sauvage, endemic to Morocco; and *L. intricatum* subsp. *intricatum*, native to Algeria, Morocco, Mauritania, South of Portugal, Southeast (SE) Spain, Italy, and the Balearic Islands. In the same work, no description about differences or similarities between those two subspecies is given. *L. intricatum* can occur in open termophilic halo-nitrophilous matorrals, next to the coastal dunes, in calcareous and saline soils [94], making it a salt tolerant plant [95]. According to the review made by these authors, this species has been considered important for the success of restoration programs in dry environments and also for hedge and wind break plant purposes. However, Marrero-Rodríguez et al. [96] studied the deforestation due to the lime industry in a specific zone of the arid island Fuerteventura (Spain), and *L. intricatum* was heavily depleted along other two species because they were systematically used for firing the kilns.

#### 2.3.1. Chemical Composition and Biological Properties

Midhat et al. [97] reported that *L. intricatum* growing near the mine of Draa Lasfar (Morocco) from which it has been extracted As, Cd, Cu, Fe, Pb and Zn, was able to accumulate high heavy metal concentrations in their shoots and roots without being affected by excessive metal contents. Regarding this observation, the authors suggest that *L. intricatum* along other plant species with the same abilities could be well suited for phytostabilization of metals-contaminated sites, in order to reduce metals dissemination by erosion or leaching, with the advantage of being well accepted by populations.

According to the review made by Yao et al. [2], seeds of *L. intricatum* have been used in helminthiasis, as a digestive, whereas the fruits have been used in eye diseases. According to Jaadan et al. [76], in the commune of Oulad Daoud Zkhanine (Morocco), the whole part of that species has been used in hair treatment. In another ethnobotanical study also made in Morocco (Tarfaya Province), Idm’hand et al. [98] reported that orally, a decoction made with leaves have been used in stomach pain and intestinal diseases, with a relative strong Fidelity Level (72%). One citation was reported by the ethnobotanical study made by Fatiha et al. [99] for the utilization of *L. intricatum* in some genitourinary ailments in the Middle Oum Rbia (Morocco).

In 2015, and for the first time, Abdennacer et al. [100] reported that polyphenols, including flavonoids, predominate in *L. intricatum* leaves collected in Tunisia, whereas anthocyanins could be detected in fruits. Nineteen phenolic compounds were isolated and fifteen were identified or tentatively characterized based on photodiode-array ultraviolet visible UV-Vis spectra, and electrospray ionisation coupled to mass spectrometry. They were: dicaffeoylquinic acid isomers, chlorognic acid, dicaffeoylputrescine, caffeoylputrescine, mono-caffeoylquinic acid, *p*-coumaroylquinic acid, feruloylquinic acid, rutin, isoquercitrin, quercetin dirhamnoside, quercitrin, kampferol rutinoside, isorhamnetin glucoside (Figure 7). Only chlorogenic acid, caffeoylputrescine, *p*-coumaroylquinic acid, feruloylquinic acid and rutin could be detected in both leaves and fruits. Although the authors reported that anthocyanins predominate in fruits, they did not present the identification of any anthocyanin. The absence of structure elucidation of the anthocyanins quantified by Abdennacer et al. [100] must be the object of further studies, since so far anthocyanin identification in fruits of *Lycium* species were only reported in *L. ruthenicum* [101].

Concerning the antioxidant activity of the fruit and leaf extracts measured through four methods, the authors verified that the leaf extracts were stronger scavenger of DPPH and ABTS or hydroxyl free radicals than the fruit extracts, which could be explained by the highest concentrations of total phenols and flavonoids. In the same year, Boulila and Bejaoui [102], described the chemical composition of the seed oil of *L. intricatum* from Northern Tunisia: linoleic acid, palmitoleic acid and erucic acid as the main fatty acids; and the hydrocarbon squalene and the triterpenic alcohols erythrodiol and uvaol. The sterolic fraction had stigmasterol, β-sitosterol and ergosterol. Later on, Bendjedou et al. [103] reported new compounds in leaf extracts of *L. intricatum* from Algeria: (1*R*,3a*R*,7a*S*)-3a,7-dimethyl-1-(*E*)-prop-1-en-1-yl-1,3a,4,7a-tetrahydroisobenzofuran-5(3H)-one; isoscopoletin; 3,4,5-trimethoxybenzyl alcohol; and (+)-isolariciresinol (Figure 8). These compounds were isolated and identified by chromatographic techniques and the structure of the first compound was also fully elucidated after ^1^H-NMR and ^13^C-NMR, MS, and electronic circular dichroism (ECD) experiments based on time-dependent density functional theory (TD-DFT).

As can be observed, the chemical composition elucidation of diverse parts of *L. intricatum* is scarce and only from Tunisia and Algeria. This means that more studies on the chemical composition of this species spread in the Mediterranean basin are needed.

#### 2.3.2. Secondary Metabolites from Endophytic Fungus

Beyond the biological activities attributed to the *L. intricatum*, there are still other compounds produced by endophytic microorganisms that also have biological attributes. In review works, the authors [104,105] reported diverse compounds isolated from endophytic fungus isolated from *L. intricatum*. Pyrenophorol, dihydropyrenophorin, 4-acetylpyrenophorol, 4-acetyldihydropyrenophorin, *cis*-dihydro-pyrenophorin, tetrahydropyrenophorin, *seco*-dihydropyrenophorin, 7-acetyl *seco*-dihydropyrenophorin, and *seco*-dihydropyrenophorin-1,4-lactone (Figure 9) were isolated from *Phoma* sp., an endophytic fungus isolated from *L. intricatum* in Spain [91]. All these compounds presented antifungal activity against *Microbotryum violaceum* (Table 2) [92]. Another review article refers other group (xanthones) of compounds (microsphaeropsones A-C) (Figure 9) isolated from *Microsphaeropsis* sp., a microorganism associated with *L. intricatum*, detected in Spain [105]. Microsphaeropsones A and C had antibacterial and algicidal activities [92,105]. However, and previously, other new components were already isolated from endophytic microorganisms in *L. intricatum* [93] and compiled in the recent review article [106]. Such compounds were a 1,4-oxazapan-7-one (microdiplactone), a hexahydroxanthone (microdiplodiasol), a 2,3-dihydrochroman-4-one (microdiplodiasone) and a 7-oxoxanthrone derivative (microdiplodiasolol) (Figure 10) isolated from *Microdiplodia* sp., with activity against *Legionella pneumophila* (Table 2).

## 3. Conclusions

The genus *Lycium* comprises 97 species and six varieties being particularly distributed in the American Continent, followed by South Africa and finally Europe and Asia. Among the *Lycium* species, *L. barbarum* has been extensively studied, particularly the berries with the purpose of finding the best conditions to enhance their shelf life or improve the berry processing for the production of snacks, drinks and other food supplements since they are considered a “superfood”. Beyond these aspects, there are also studies focused on the chemical composition, biological activities, and clinical applications. In spite of more than 355 compounds having been isolated and identified from the *Lycium* genus distributed into diverse compound classes (polysaccharides, glycerogalactolipids, phenylpropanoids, coumarins, lignans, flavonoids, amides, alkaloids, anthraquinones, organic acids, terpenoids, sterols, steroids and their derivatives, and peptides), they were mainly described for *L. barbarum* and *L. chinense* [36,39,107]. In contrast, significantly fewer studies have been focused on other *Lycium* species such as *L. europaeum*, *L. intricatum*, *L. schweinfurthii* and *L. infaustum*.

The present review concluded that there is little research on chemical composition and biological properties of *L. infaustum* (at least published in scientific articles). Concerning *L. intricatum*, only very few compounds (nineteen polyphenols, fatty acids, terpenes and phytosterols) were identified in samples from Algeria and Tunisia, in spite of this plant occurring in other Mediterranean countries, as well as in Portugal, Mauritania, Saudi Arabia, and Mexico. However, the interest by this species seems to be increasing, since in the last ten years scientific publications regarding their chemical and biological properties have started to become available. For example, in 2022, three new compounds, 3,4,5-trimethoxybenzyl alcohol, (+)-isolariciresinol and [(1R,3aR,7aS)-3a,7-dimethyl-1-(*E*)-prop-1-en-1-yl-1,3a,4,7a–tetrahydroisobenzofuran-5(3H)-one] were identified in leaves of *L. intricatum* from Algeria, not yet described in other *Lycium* species, being the former a completely new compound, never previously reported. In Morocco, ethnobotanical studies have demonstrated the application of *L. intricatum* in hair treatment, stomach pain and intestinal diseases.

Very few works about the chemical composition and biological activities of *L. schweinfurthii* can be found, with most research focused on Egyptian plants, although this species can also be found in other places, such as Portugal, Spain, Israel, Morocco, Greece, Algeria, Tunisia, Mauritania, and Cyprus. Diverse classes of compounds have been identified, which are within those previously summarized for *Lycium* species [107]. However, in samples of Egyptian origin, a new compound of a natural source was identified, 3-methoxy-4-*O*-β-D-glucopyranosyl-methylbenzoate, and compounds not yet found in *Lycium* species [vaginatin, (*E*)-docosanoyl ferulate and (*Z*)-docosanoyl ferulate, methyl-α-D-fructofuranoside and meliasendanin D]. The above-mentioned new compound was shown to be a potent inhibitor of α-glucosidase activity, along with diosmetin, luteolin and quercetin. Nevertheless, interest in the antioxidant activity of *L. schweinfurthii* has started to emerge since the production of in vitro plants and cell suspension cultures that were developed with the purpose of obtaining metabolites with antioxidant properties with the advantage of being independent of the field conditions.

Through a review made by Wannes and Tounsi [64], in 2021, it was possible to determine that only 30 constituents could be identified in *L. europaeum*, distributed by the same classes already reported for *Lycium barbarum* [107]. Diverse biological attributes were also reported in that compilation, including: antioxidant, anti-inflammatory, antinociceptive, hypoglycemic, hypolipidemic, hepatoprotective, nephroprotective and cytotoxic activities.

## 4. Future Trend

The present compilation intends to call attention to of the appropriate interlocutors to the need of studying some species, such as *L. europaeum*, *L. intricatum*, *L. infaustum* and *L. schweinfurthii*, which are forgotten because other more competitive and assertive markets are managing the introduction of their products on the global market. This review can be the trigger for the beginning of more in-depth studies on these species with the aim of knowing if they can have the same uses of *L. barbarum* or even new applications. The evaluation of chemical composition, biological, nutritional, or pharmacological properties of those species must be unraveled. This approach can contribute to a reduction in economic and energy costs, that is, a more environmentally friendly production, since those species could be produced in their native habitats. Since *L. europaeum*, *L. intricatum*, *L. infaustum* and *L. schweinfurthii* generally occur in impoverished areas, the culture and transformation of these species products could contribute to the sustained enrichment of the populations living in those zones. However, it is also important to avoid the cultivation of the most promising species or their genotypes for the sole purposes of the industrial production of dried fruit, as has occurred with the fruits of *L. barbarum*. Other applications must be thought of and considered, always bearing in mind the preservation of all species and varieties, in order to save biodiversity.

## Figures and Tables

**Figure 1 metabolites-12-01265-f001:**
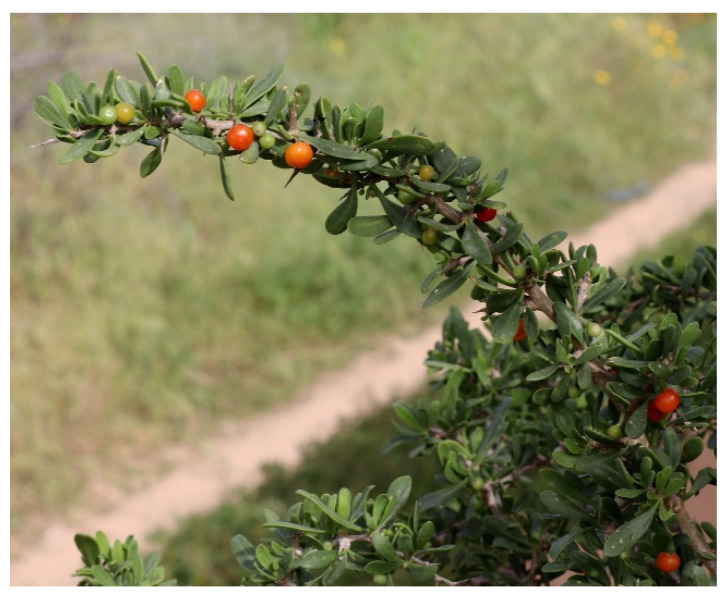
*Lycium europaeum* L. (Source: https://www.plantarium.ru/lang/en/page/image/id/560323.html) (accessed on 24 September 2022).

**Figure 2 metabolites-12-01265-f002:**
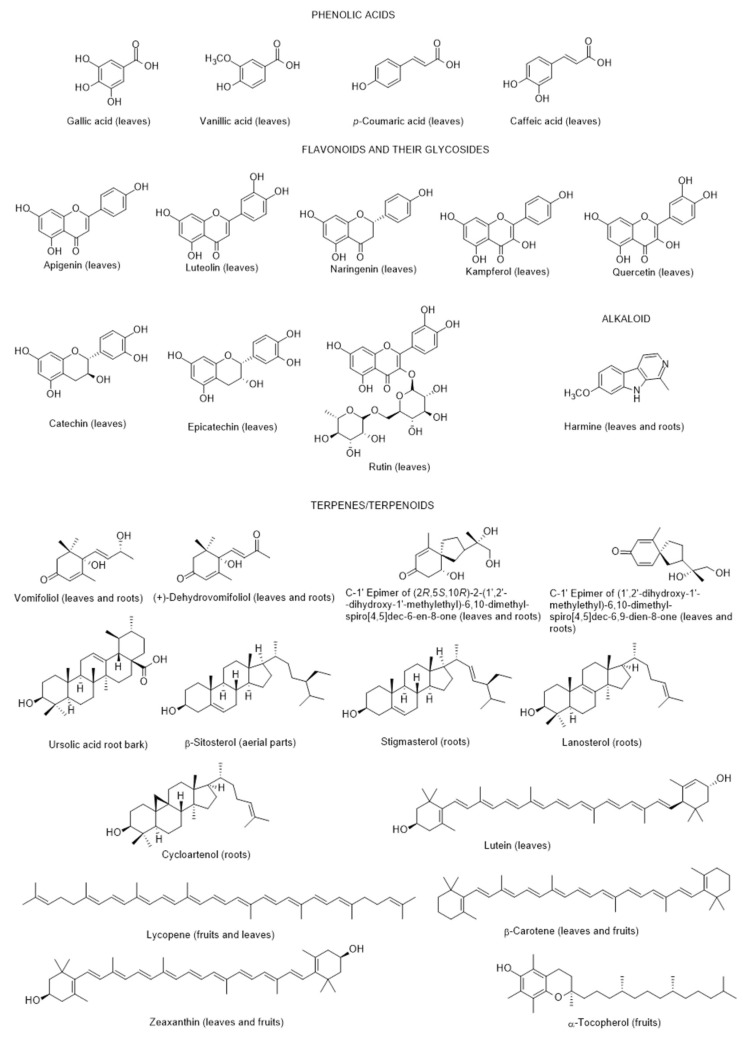
Metabolites found in different parts of *Lycium europaeum*.

**Figure 3 metabolites-12-01265-f003:**
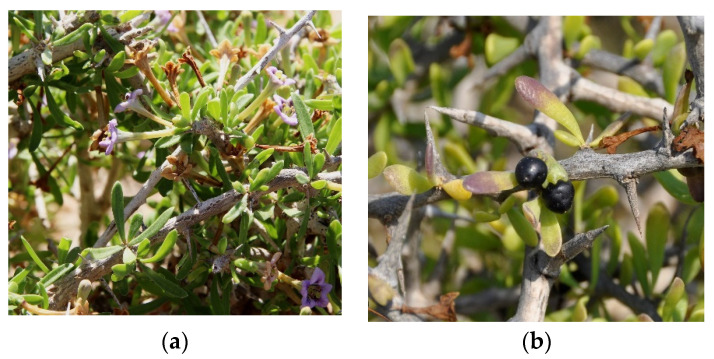
*Lycium schweinfurthii* Dammer: (**a**) general aspects of the aerial parts (source: https://www.plantarium.ru/lang/en/page/image/id/652684.html) (accessed on 24 September 2022); and (**b**) general aspects of fruits (source: https://www.plantarium.ru/lang/en/page/image/id/627979.html).

**Figure 4 metabolites-12-01265-f004:**
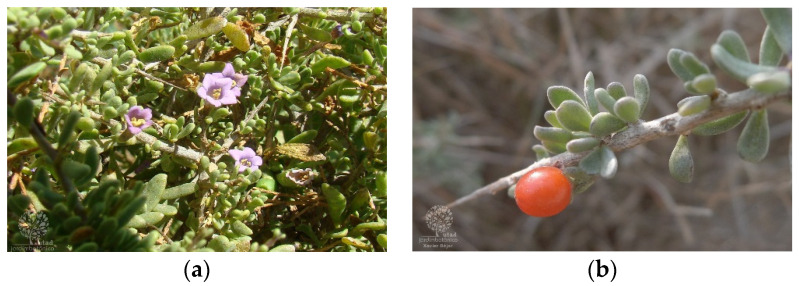
*Lycium intricatum* Boiss: (**a**) general aspects of the aerial parts (source: https://jb.utad.pt/multimedia/10921 (accessed on 24 September 2022); Imagem da espécie Lycium intricatum do Botânico UTAD, Flora Digital de Portugal.); and (**b**) general aspects of fruits and leaves (https://jb.utad.pt/multimedia/15805); Imagem da espécie Lycium intricatum por Xavier Béjar do Jardim Botânico UTAD, Flora Digital de Portugal.) https://www.plantarium.ru/lang/en/page/image/id/627979.html).

**Figure 5 metabolites-12-01265-f005:**
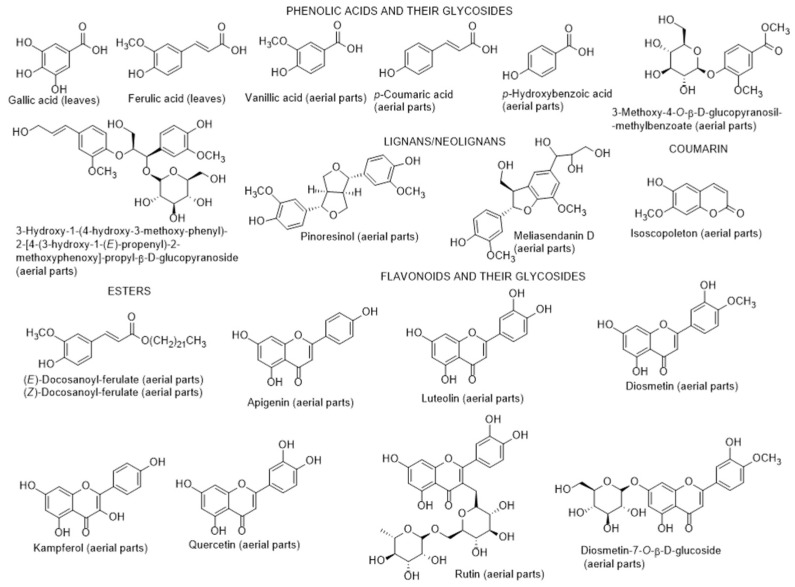
Metabolites found in different parts of *Lycium schweinfurthii*.

**Figure 6 metabolites-12-01265-f006:**
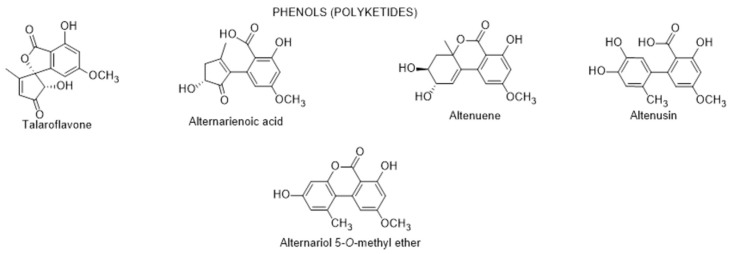
Metabolites isolated from the endophytic *Alternaria* sp. isolated from the leaves of *L. schweinfurthii*.

**Figure 7 metabolites-12-01265-f007:**
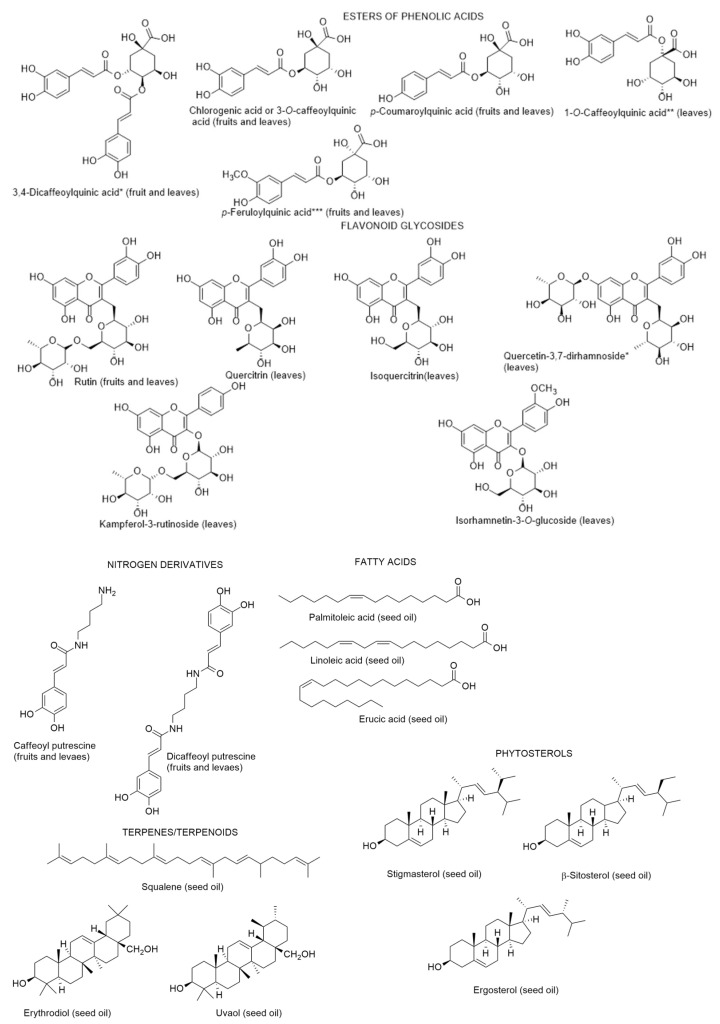
Metabolites found in different parts of *Lycium intricatum*. *, **, *** The isomers were not reported by the authors [99], therefore, the compounds represented are examples and not the isomers identified.

**Figure 8 metabolites-12-01265-f008:**
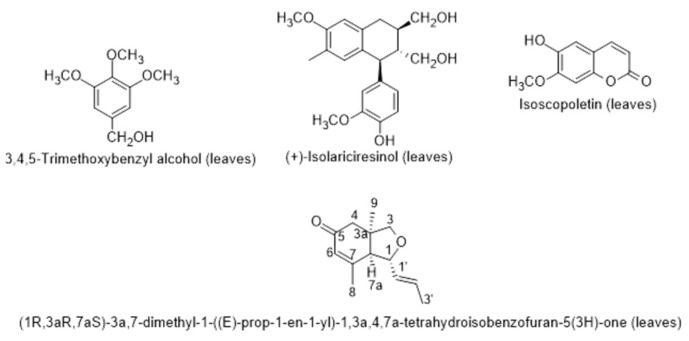
Metabolites found in leaf extracts of *Lycium intricatum* from Algeria [101].

**Figure 9 metabolites-12-01265-f009:**
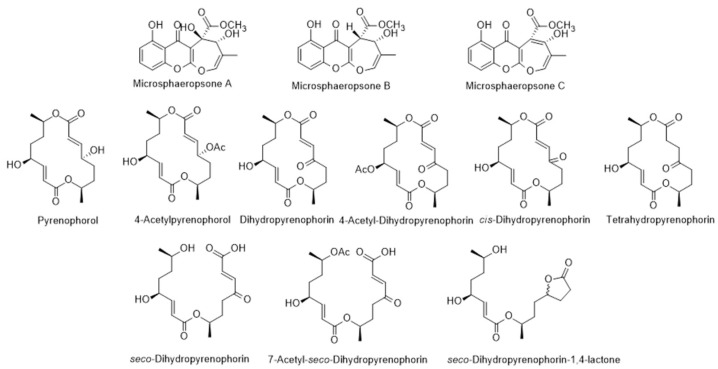
Metabolites isolated from endophytic *Microsphaeropsis* sp. (the first three compounds) and *Phoma* sp. isolated from *L. intricatum*.

**Figure 10 metabolites-12-01265-f010:**
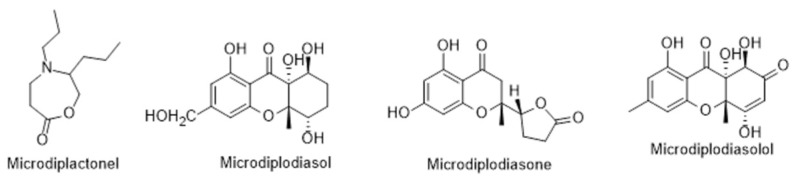
Metabolites isolated from endophytic *Microdiplodia* sp. isolated from *L. intricatum*.

**Table 1 metabolites-12-01265-t001:** Biological activities of isolated compounds of the aerial parts of *Lycium schweinfurthii* (adapted from [83,84]).

Compound	Cell Line	Assay	IC_50_ (μM)	Reference
Diosmetin	NT	Inhibition of α-glucosidase	23.55	[83]
Quercetin	NT	Inhibition of α-glucosidase	21.38	[83]
Diosmetin	NT	Inhibition of α-glucosidase	7.12	[83]
Diosmetin-7-*O*-β-D-glucoside	NT	Inhibition of α-glucosidase	168.85	[83]
3-Methoxy-4-*O*-β-D-glucopyranosyl-methylbenzoate	NT	Inhibition of α-glucosidase	22.76	[83]
3-Hydroxy-1-(4-hydroxy-3-methoxyphenyl)-2[4-(3-hydroxy-1-(*E*)-propenyl)-2-methoxyphenoxy]-propyl-β-D-glucopyranoside	NT		181.39	[83]
Vaginatin	1/2/3	Cytotoxicity	25.65/82.91/69.18	[84]
11*S*-Methoxy-11,12-dihydrophytuberin	1/2/3	Cytotoxicity	92.09/>100/>100	[84]
(*E*)-Docosanoyl ferulate	1/2/3	Cytotoxicity	73.49/>100/>100	[84]
(*Z*)-Docosanoyl ferulate	1/2/3	Cytotoxicity	>100/>100>100	[84]
β-Sitosterol	1/2/3	Cytotoxicity	NT/>100/>100	[84]
β-Sitosterol-3-*O*-β-D-glucoside	1/2/3	Cytotoxicity	53.19/>100>96.19	[84]
9*S*-Methoxy-benzocyclononan-7-one	1/2/3	Cytotoxicity	47.34/>100/>100	[84]
Isoscopoletin	1/2/3	Cytotoxicity	31.90/>100/>100	[84]
Pinoresinol	1/2/3	Cytotoxicity	34.57/>100/75.76	[84]
*trans*-Cinnamoyl tyramine	1/2/3	Cytotoxicity	45.58/>100/>100	[84]
*trans*-Ferulyl tyramine	1/2/3	Cytotoxicity	58.76/>100/90.26	[84]
*trans*-Ferulyl-3-methoxytyramine	1/2/3	Cytotoxicity	7.14/63.39/51.74	[84]
Diosmetin	1/2/3	Cytotoxicity	42.17/40.24/40.57	[84]
Apigenin	1/2/3	Cytotoxicity	54.49/>100/84.32	[84]
Luteolin	1/2/3	Cytotoxicity	NT/>100/>100	[84]
*p*-Hydroxybenzoic acid	1/2/3	Cytotoxicity	NT/>100/>100	[84]
Vanillic acid	1/2/3	Cytotoxicity	9.55/62.35/61.68	[84]
Kampferol	1/2/3	Cytotoxicity	49.10/>100/>100	[84]
Quercetin	1/2/3	Cytotoxicity	NT/NT/NT	[84]
Methyl-α-D-fructofuranoside	1/2/3	Cytotoxicity	NT/NT/NT	[84]
Meliasendanin D	1/2/3	Cytotoxicity	>100/>100/>100	[84]
*trans*-Ferulic acid	1/2/3	Cytotoxicity	NT/>100/>100	[84]
*p*-Coumaric acid	1/2/3	Cytotoxicity	NT/>100/>100	[84]
Diosmetin-7-*O*-β-D-glucoside	1/2/3	Cytotoxicity	NT/>100/>100	[84]
Rutin	1/2/3	Cytotoxicity	89.47//>100/>100	[84]
3-Methoxy-4-*O*-β-D-glucopyranosyl-methylbenzoate	1/2/3	Cytotoxicity	NT/>100/86.76	[84]
Gallic acid	1/2/3	Cytotoxicity	11.09/>100/75.92	[84]
3-Hydroxy-1-(4-hydroxy-3-methoxyphenyl)-2[4-(3-hydroxy-1-(*E*)-propenyl)-2-methoxyphenoxy]-propyl-β-D-glucopyranoside	1/2/3	Cytotoxicity	NT/>100/>100	[84]

NT: not tested; 1: cancer cell lines from skin (G-361); 2: cancer cell lines from colon (HCT-116); 3: cancer cell lines from colon (Caco-2); IC_50_: the concentration of the tested compound (μM) that inhibited the α-glucosidase activity by 50%; or the concentration of the tested compound (μM) that decreased the number of viable cells by 50%.

**Table 2 metabolites-12-01265-t002:** Biological properties found for some compounds isolated from endophytic fungi found in *L. schweinfurthii* and *L. intricatum* (adapted from [89,91,92,93].

	Inhibition of Enzymatic Activity		
Compound/Endophytic Fungus	α-Glucosidase IC_50_ (μM)	Pancreatic Lipase IC_50_ (μM)	References
*Alternaria* sp.	*Lycium schweinfurthii*		
Talaroflavone	>300	73.82	[89]
Alternarienoic acid	7.95	20.82	[89]
Altenuene	40.38	3.18	[89]
Altenusin,	46.14	21.46	[89]
Alternariol	179.88	56.85	[89]
Alternariol-5-*O*-methyl ether	236.25	7.60	[89]
**Compound/Endophytic Fungus**	**Antimicrobial Activity**		
*Phoma* sp.	*Lycium intricatum*		
	*Microbotryum violaceum* *		
Pyrenophorol	**7**		[91]
Dihydropyrenophorin	**7**		[91]
4-Acetylpyrenophorol 4-Acetyldihydropyrenophorin	10 7		[91] [91]
*cis*-Dihydro-pyrenophorin	7		[91]
Tetrahydropyrenophorin	10		[91]
*seco*-Dihydropyrenophorin	10		[91]
7-acetyl *seco*-dihydropyrenophorin	9		[91]
*seco*-Dihydropyrenophorin-1,4-lactone	**7**		[91]
*Microsphaeropsis* sp.	*Lycium intricatum*		
	*Escherichia coli **	*Bacillus megaterium **	
Microsphaeropsone A	8	10	[92]
Microsphaeropsone C	6	10	[92]
*Microdiplodia* sp.	*Lycium intricatum*		
	*Microbotryum violaceum* *	*Legionella pneumophila ***	
Microdiplodiasol	7	0.125–0.25	[93]
Microdiplodiasone	0	1.0	[93]
Microdiplodiasolol	6	1.0	[93]

* Radius of zone of inhibition measured in mm; ** MIC (minimum inhibitory concentration) in mg/mL.

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
