# Peer review of "Chemical and Biological Properties of Three Poorly Studied Species of Lycium Genus—Short Review"

_metabolites, 2022, doi:10.3390/metabo12121265_

Round 1
Reviewer 1 Report
1- Page 1 line 1: the title suggest; Chemical and Biological properties of three 2 poorly studied species of the Lycium genus- Short review instead of Short review on chemical and biological properties of three 2 poorly studied species of the Lycium genus.
2- Page 1 lines 10-16: the abstract is not enough need more details.
3- Page 1 line 17 : in Keywords the chemical and biological properties of Lycium genus should be added
4- Page3 lines 148-150: the aime of the work is not clear and need improvement
5- Page 14 lines 527-591 : Conclusions is so long and need to summarized
6- The authors should added the future trend for their subject
Author Response
Dear Dr. Praewpran Chavanon,
I thank you for your e-mail of 24 November 2022 with the Decision Letter on Manuscript ID: metabolites-2079078, titled: Chemical and biological properties of three poorly studied species of Lycium genus- Short review.
I have read it carefully and I do understand most of the referee’s comments. Please find below our reply and comments addressing each point raised by the reviewer. All corrections, additions and changes performed in the MS and they are marked with yellow colour.
Reviewer 1
Comments and Suggestions for Authors
- Page 1 line 1: the title suggest; Chemical and Biological properties of three 2 poorly studied species of the Lycium genus- Short review instead of Short review on chemical and biological properties of three 2 poorly studied species of the Lycium genus.
Thank you for the suggestion.
- Page 1 lines 10-16: the abstract is not enough need more details.
More details were introduced in the Abstract, hoping that this will significantly improve it.
- Page 1 line 17 : in Keywords the chemical and biological properties of Lycium genus should be added
They were added as suggested.
- Page3 lines 148-150: the aime of the work is not clear and need improvement
Whereas there are many scientific studies involving the chemical composition, biological properties, conservation and drying methods of the L. barbarum fruit, resulting in more than 20 review articles in indexed scientific journals, in the first 11 months of 2022, the same cannot be observed for other species, such as L. europaeum, L. intricatum and L. schweinfurthii. This review, by compiling the little information that exists on 3 species of Lycium, can contribute to arouse interest in such species as possible sources of food, food supplements or even medicines.
- Page 14 lines 527-591 : Conclusions is so long and need to summarized
They were summarized
6- The authors should added the future trend for their subject
It was introduced.
Reviewer 2
Comments and Suggestions for Authors
This manuscript aimed to summarize the chemical and biological properties of three species of Lycium genus. However, the author just mainly introduces the chemical composition in Lycium genus based on the previous review paper. Furthermore, the language of this manuscript is not good. Too many long and complex sentences were used, and the grammar errors were often found. It is very difficult to understand the expression for many sentences in the manuscript. Some detailed comments were listed as follows.
L3, delete “the”..
The title was modified as suggested by reviewer 1 and in this new version, the term “the” was deleted.
L14-16, please rewrite these two sentences.
The fisrt sentence was re-writtem and the last sentence was deleted.
L194-199, too long sentence.
It was reformulated.
L267-269, too long sentence.
It was reformulated.
L277, will be.
It was introduced.
L308, By using chromatographic techniques, five compounds including quercetin, kaempferol, apigenin, gallic, and ferulic acids, were separated from leaves.
The sentence was introduced. Thank you for the correction.
L471, please revised the compound structure.
The squalene structure was revised.
L495-498, too long sentence.
It was reformulated.
L530-535, too long sentence.
It was reformulated.
L538, distributed into.
It was corrected.
L542, Much less studies focused on….
It was modified: In contrast, much less studies have been focused on other Lycium species such as L. europaeum, L. intricatum, L. schweinfurthii and L. infaustum.
L582-586, too long and complex sentence.
It was reformulated.
L590, delete the first “and”.
The sentence was deleted.
Reviewer 3
Comments and Suggestions for Authors
- The paper reports the chemical and biological aspects of species from the Lycium genus. This review study intends is interesting to trigger the interest for these species. But, there is an important reference on Lycium genus which was not cited: Fukuda et al 2001 (Phylogeny and biogeography of the genus Lycium (Solanaceae): inferences from chloroplast DNA sequences). Please use this reference to support this review.
It was introduced. I apologize for this failure.
- the paper is difficult to read because of the huge information that could be structured in more sections for each species as distribution, composition and activities …
Some subsections were introduced.
- please correct « varities » (line 28)
It was corrected.
- Please add reference to this sentence: “The polysaccharide complex with … nosyluronic acid residues” (line 120-124)
The references are 37, 40 and 41.
- Please reformulate the last paragraph if the introduction. It seems confusing when you write “it is possible to find a large number of review papers that have..” that means there no need to review papers on Lycium. To be checked and rephrased if necessary ..
It was reformulated.
- Figures of compounds found in lycium studied species could be classified in tables, if possible, with biological activities in order to be related to major topic of the paper. The chemical structure of each compounds is known, the list of names is enough. Please summarize the compounds in tables with names, activities and references (it could be more informative).
Thank you for the suggestion. I have introduced two Tables regarding the biological properties found, but I did not remove the figures.
- In conclusion section, there is repetition of some sentences already mentioned in the introduction. So it should be reduced to the main new information..
It was reformulated.
Reviewer 4
- I have read the entire manuscript and my initial comment is that manuscript is well-written. I have significant concerns about the grammar and vocabulary of the manuscript; therefore, improvement of the language is highly needed.
- The structure of the abstract should be improved, as well as the lack of several aspects that should be included in this section. Most of the abstracts contain confusing and uninformative sentences. Please give more precise objectives here (such as in the Abstract).
It was reformulated.
- Introduction grammatical issues appear to be most prevalent in the introduction, making for very confusing reading. Further, the introduction is short but has no clear thread.
It was reformulated, mainly the objectives.
- The conclusion section must be improved.
It was reformulated.
- References: shall have to correct the whole References according to the ”Instructions for the Authors”, e.g. title should not be in italics, the Journal name is in italics, and the author shall have to use the abbreviated name Journals cited the year must be bold, the scientific name must be italics etc. Please check all references carefully.
I need help, because I think I have the references according to the ”Instructions for the Authors”.
I hope that I have adequately addressed the reviewer remarks and questions, and that the manuscript is now suitable for publication.
Yours sincerely,
Maria da Graça Miguel
Reviewer 2 Report
This manuscript aimed to summarize the chemical and biological properties of three species of Lycium genus. However, the author just mainly introduces the chemical composition in Lycium genus based on the previous review paper. Furthermore, the language of this manuscript is not good. Too many long and complex sentences were used, and the grammar errors were often found. It is very difficult to understand the expression for many sentences in the manuscript. Some detailed comments were listed as follows.
L3, delete “the”..
L14-16, please rewrite these two sentences.
L194-199, too long sentence.
L267-269, too long sentence.
L277, will be.
L308, By using chromatographic techniques, five compounds including quercetin, kaempferol, apigenin, gallic, and ferulic acids, were separated from leaves.
L471, please revised the compound structure.
L495-498, too long sentence.
L530-535, too long sentence.
L538, distributed into.
L542, Much less studies focused on….
L582-586, too long and complex sentence.
L590, delete the first “and”.
Author Response

(The authors gave the same response as above.)

Reviewer 3 Report
- The paper reports the chemical and biological aspects of species from the Lycium genus. This review study intends is interesting to trigger the interest for these species. But, there is an important reference on Lycium genus which was not cited: Fukuda et al 2001 (Phylogeny and biogeography of the genus Lycium (Solanaceae): inferences from chloroplast DNA sequences). Please use this reference to support this review.
- the paper is difficult to read because of the huge information that could be structured in more sections for each species as distribution, composition and activities …
- please correct « varities » (line 28)
- Please add reference to this sentence: “The polysaccharide complex with … nosyluronic acid residues” (line 120-124)
- Please reformulate the last paragraph if the introduction. It seems confusing when you write “it is possible to find a large number of review papers that have..” that means there no need to review papers on Lycium. To be checked and rephrased if necessary ..
- Figures of compounds found in lycium studied species could be classified in tables, if possible, with biological activities in order to be related to major topic of the paper. The chemical structure of each compounds is known, the list of names is enough. Please summarize the compounds in tables with names, activities and references (it could be more informative).
- In conclusion section, there is repetition of some sentences already mentioned in the introduction. So it should be reduced to the main new information..
Author Response

(The authors gave the same response as above.)

Reviewer 4 Report
To,
The Editor,
Metabolites, MDPI,
Manuscript ID: metabolites-2079078
Subject: Submission of comments of the manuscript in “Metabolites"
Dear Editor Metabolites, MDPI,
Thank you very much for the invitation to consider a potential reviewer for the manuscript (ID: metabolites-2079078). My comments responses are furnished below as per each reviewer’s comments.
In the reviewed manuscript, authors analyzed the chemical composition and the biological properties of L. europaeum, L. intricatum and L. schweinfurthii, found in diverse places, particularly in the Mediterranean region. This study permitted to conclude that currently very few works have been done concerning these species in perfect contrast to the observed for L. barbarum. This review intends to trigger the interest for these species, augmenting their value and in doing so contributing for biodiversity. In general, the manuscript represents a very big piece of information. Therefore, it might be conditionally accepted subject to minor revision. Authors have to improve their manuscripts with many non-clear meanings, inaccuracies and inconsistencies, and the authors need to address the following issues before it can be accepted for publication.
1. I have read the entire manuscript and my initial comment is that manuscript is well-written. I have significant concerns about the grammar and vocabulary of the manuscript; therefore, improvement of the language is highly needed.
2. The structure of the abstract should be improved, as well as the lack of several aspects that should be included in this section. Most of the abstracts contain confusing and uninformative sentences. Please give more precise objectives here (such as in the Abstract).
3. Introduction grammatical issues appear to be most prevalent in the introduction, making for very confusing reading. Further, the introduction is short but has no clear thread.
4. The conclusion section must be improved.
5. References: shall have to correct the whole References according to the ”Instructions for the Authors”, e.g. title should not be in italics, the Journal name is in italics, and the author shall have to use the abbreviated name Journals cited the year must be bold, the scientific name must be italics etc. Please check all references carefully.
Author Response

(The authors gave the same response as above.)

Round 2
Reviewer 2 Report
The manuscript has been well improved. I recommend that it can be accepted.
Author Response
Dear Assistant Editor,
Dr. Praewpran Chavanon,
Date: 07/12/2022
Subject: Metabolites Decision Letter Reply
Dear Dr. Praewpran Chavanon,
I thank you for your e-mail of 07 November 2022 with the Decision Letter on Manuscript ID: metabolites-2079078, titled: Chemical and biological properties of three poorly studied species of Lycium genus- Short review.
I have read it carefully and I do understand most of the referee’s comments. Please find below our reply and comments addressing each point raised by the reviewer. All corrections, additions and changes performed in the MS and they are marked with gray colour.
Reviewer 2
Comments and Suggestions for Authors
The manuscript has been well improved. I recommend that it can be accepted.
Thank you for you decision.
Reviewer 3
Comments and Suggestions for Authors
In general, the authors adreess the majority of referees'comments, but some of them are missed. the authors are required to revise the following points:
- The section 2.1. contains only one subsection “2.1.1. Chemical composition and biological properties”. The other subsection were missed. Please check.
I have introduced the section 2.1.2.: Fungal infection.
- Many long and complex sentences were used with grammar errors . It is very difficult to understand the meaning of many sentences in the manuscript.English revision is highly required.
I have done some corrections. Unfortunately I am not able to do better.
I hope that I have adequately addressed the reviewer remarks and questions, and that the manuscript is now suitable for publication.
Yours sincerely,
Maria da Graça Miguel
Reviewer 3 Report
In general, the authors adreess the majority of referees'comments, but some of them are missed. the authors are required to revise the following points:
- The section 2.1. contains only one subsection “2.1.1. Chemical composition and biological properties”. The other subsection were missed. Please check.
- Many long and complex sentences were used with grammar errors . It is very difficult to understand the meaning of many sentences in the manuscript.English revision is highly required.
Author Response

(The authors gave the same response as above.)
